# PROCEEDINGS A

mathematical modelling, computational mathematics, applied mathematics

COVID-19, economy, infection, agent-based model

**Author for correspondence:**
Takeshi Kano
e-mail: tkano@riec.tohoku.ac.jp

# An agent-based model of the interrelation between the COVID-19 outbreak and economic activities

Takeshi Kano[1], Kotaro Yasui[1,2], Taishi Mikami[1], Munehiro Asally[3,4,5] and Akio Ishiguro[1]

[1]Research Institute of Electrical Communication, Tohoku University, 2-1-1 Katahira, Aoba-ku, and [2]Frontier Research Institute for Interdisciplinary Sciences, Tohoku University, Aramaki aza Aoba 6-3, Aoba-ku, Sendai 980-8578, Japan
[3]School of Life Sciences, [4]Warwick Integrative Synthetic Biology Centre, [5]Bio-Electrical Engineering Innovation Hub, University of Warwick, Coventry CV4 7AL, UK

TK, 0000-0002-2033-4695

As of July 2020, COVID-19 caused by SARS-COV-2 is spreading worldwide, causing severe economic damage. While minimizing human contact is effective in managing outbreaks, it causes severe economic losses. Strategies to solve this dilemma by considering the interrelation between the spread of the virus and economic activities are urgently needed to mitigate the health and economic damage. Here, we propose an abstract agent-based model of the COVID-19 outbreak that accounts for economic activities. The computational simulation of the model recapitulates the trade-off between the health and economic damage associated with voluntary restraint measures. Based on the simulation results, we discuss how the macroscopic dynamics of infection and economics emerge from individuals' behaviours. We believe our model can serve as a platform for discussing solutions to the above-mentioned dilemma.

## 1. Introduction

COVID-19, caused by SARS-COV-2, was first reported in Wuhan in December 2019 [1–6]; it spread globally,

causing severe health and economic damage. By 29 July 2020, over 16 million people had tested positive and 660 thousand people had died due to COVID-19 [7]. From the economic perspective, many people suffered losses and lost their jobs [8]. Therefore, we find a dilemma between mitigating the spread of COVID-19 and reducing economic losses, that is, strategies to decrease contact between humans, such as voluntary restraint and social distancing, hinder normal economic activities. Hence, solving this dilemma by finding strategies to mitigate the outbreak while minimizing economic losses is an urgent issue.

Many mathematical models of the spread of infectious diseases have been proposed using differential equations [9–12] and agent-based models [13–15], and several studies focus on the spread of COVID-19 in various contexts [16–26]. These studies model only the spread of infectious diseases and do not describe economic activities mathematically. Other studies focused on the economic impacts of COVID-19 [27–31], some of which considered economic processes in detail and predicted economic impacts under realistic assumptions [27–29]; thus, they are complex and make it difficult to capture the essence of the interrelation between the spread of the virus and economic activities. Other studies estimated the economic impact using a simple model based on differential equations [30,31]. While these models describe the economic effects at the population level, the effects of individuals' behaviours on the macroscopic dynamics of infection and the economy remain unclear. A definitive mathematical model that captures the essential mechanism of the interrelation between the spread of the virus and economic activities at an individual level could fill this gap.

In this study, we propose a simple mathematical model that considers both COVID-19 infections and economic activities. Our aim is to extract the essence of the relationship between the spread of COVID-19 and economic activities, rather than making a quantitative and accurate prediction of infection and economic outcomes. Moreover, we are interested in how the macroscopic dynamics of infection and economic effects emerge from individuals' behaviours, rather than a coarse macroscopic description. Hence, we propose a highly abstract and simple agent-based model without employing detailed and realistic assumptions. More specifically, we propose a cellular automaton model in which mobile agents with internal states regarding infection and the economy interact with others and update their internal states. Through simulations, we demonstrate how the health and economic states evolve depending on several parameters. Based on the results, we discuss the effect of voluntary restraint.

## 2. Model

### (a) Overview of the proposed model

We consider a cellular automaton model in which hexagonal cells are aligned regularly on a two-dimensional plane (figure 1) with $N$ agents. Each agent has a home cell and can stay at home or move to its adjacent cell at each time step. Each agent has a health state $State_i$ and money $M_i$, which are updated through interaction with other agents. Agents die when they do not recover after infection or when their money becomes zero.

Agents have their own businesses. There are several types of businesses that range from selling commodities to selling non-essentials. Each agent chooses one type among them and does not change temporarily. Each agent has a demand for goods. When the demand exceeds a certain threshold, the agent goes out to buy them. The agent pays money when it reaches the home of an agent who sells the goods (figure 1). We assume that agent $i$ sells goods to agent $j$ when agent $j$ visits the home cell of agent $i$. The model does not include the prime costs of goods. When agents perceive that infection is spreading around themselves, they tend not to demand non-essentials, while they demand commodities as usual. In the following subsections, we describe the details of the proposed model.

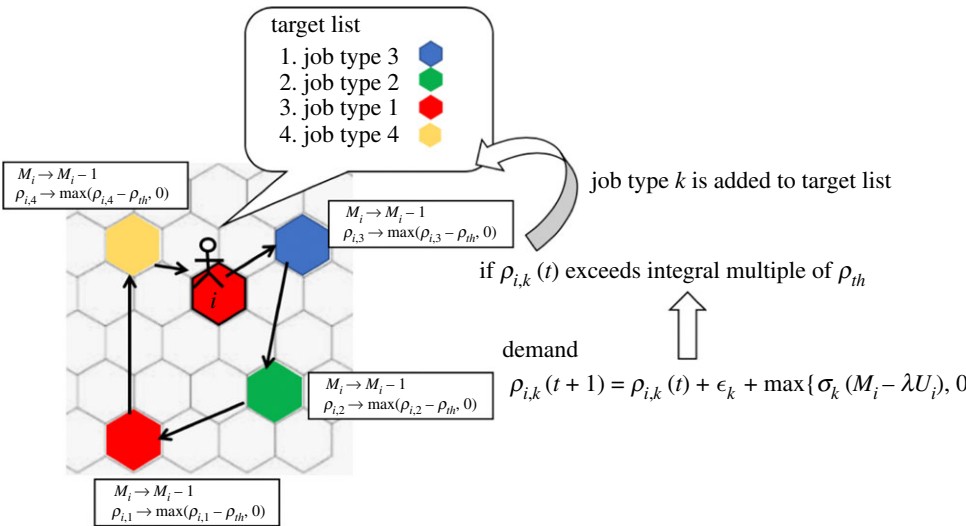

**Figure 1.** Outline of the proposed model. When $\rho_{i,k}(t)$ exceeds the integral multiple of $\rho_{th}$, job type $k$ is added to the bottom of the target list of agent $i$. Agent $i$ visits the home of the nearest agent among the agents whose job type corresponds to the top of the target list. When the agent reaches the home, the agent pays a unit amount of money to buy goods. Then because the demand of agent $i$ is satisfied, $\rho_{i,k}$ decreases by $\rho_{th}$. The top item in the target list is removed, and the other items in the target list move up. Then, agent $i$ visits the next target. When there are no items in the target list, agent $i$ returns home and remains there. (Online version in colour.)

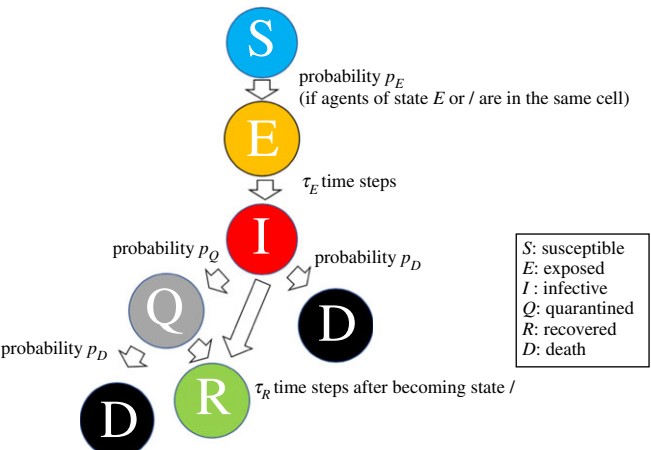

**Figure 2.** Rule for the transition of the health state. (Online version in colour.)

## (b) Model of infection

We model the spread of COVID-19 by drawing inspiration from a spatial susceptible-exposed-infectious-recovered (SEIR) model [15]. Each agent has a health state $State_i$, which has a susceptibility state ($S$), an asymptomatic infection state ($E$), a symptomatic infection state ($I$), a quarantine state ($Q$), a recovered state ($R$) and a death state ($D$).

The state changes according to the rule in figure 2. When an agent with state $S$ is in a cell occupied by an agent with state $E$ or $I$, $State_i$ changes from $S$ to $E$ with probability $p_E$ at every time step. Here, although the existing SEIR model [15] assumes that only agents with state $I$ have infectability, we assume that agents with states $E$ or $I$ have infectability because asymptomatic COVID-19 patients have infectability [32]. When $\tau_E$ time steps pass after the transition from

state $S$ to $E$, the state changes to state $I$. Agents with state $I$ change to state $Q$ and $D$ with probability $p_D$ and $p_Q$, respectively, at every time step. Agents with state $Q$, which represents hospitalized patients, do not have infectability and stop any economic activity described in the next subsection. Agents with state $Q$ also die with probability $p_D$ at every time step. Regardless of whether quarantined or not, the state changes to $R$ when $\tau_R$ time steps pass after the transition from state $E$ to $I$.

## (c) Model of economic activity

Each agent has money $M_i$, which increases and decreases when other agents buy agent $i$'s goods and when agent $i$ buys goods, respectively. We assume that the price of all goods is one unit, which does not change temporarily. Namely, when agent $i$ sells a good to agent $j$, $M_i$ and $M_j$ increase and decrease by one, respectively. In addition to buying and selling, high-income people pay high taxes, while low-income people receive public assistance in real societies. Although these policies are complex, we simply assume that the money of each agent increases and decreases with these policies when their economic levels are below and above the average level, respectively. Thus, we describe the time evolution of $M_i$ as

$$M_i(t+1) = M_i(t) + q_{in,i}(t) - q_{out,i}(t) + K(\hat{M} - M_i(t)), \tag{2.1}$$

where $t$ is a time step, $K$ and $\hat{M}$ are positive constants and $q_{in,i}(t)$ and $q_{out,i}(t)$ denote the amount of money that increases or decreases via selling and buying, respectively. The fourth term on the right-hand side represents the effect of income redistribution: high-income people pay high taxes and low-income people receive public assistance. The parameter $K$ represents the extent of the income redistribution policy. Agent $i$ dies when $M_i(t)$ becomes zero.

Each agent chooses from one of $m$ job types, which does not change temporarily. Agent $i$'s demand for goods produced by job types $k$ ($i = 1, 2, \ldots, N$ and $k = 1, 2, \ldots, m$) are denoted by $\rho_{i,k}$. The time evolution of $\rho_{i,k}$ is

$$\rho_{i,k}(t+1) = \rho_{i,k}(t) + \epsilon_k + \max\{\sigma_k(M_i - \lambda U_i), 0\}, \tag{2.2}$$

where $\epsilon_k$, $\sigma_k$ and $\lambda$ are positive constants, and $U_i$ quantifies agent $i$'s perception of 'the extent to which infection is spreading around itself', which we formulate below. Parameter $\epsilon_k$ represents the rate of increase in the demand which does not depend on the money agent $i$ owns or the degree of the outbreak. By contrast, $\sigma_k$ represents the rate of increase in demand which is affected by the money agent $i$ owns or the degree of the outbreak. The third term on the right-hand side of equation (2.2) means that the demand increases rapidly when agent $i$ has a lot of money, though the increase stops under the outbreak. Parameter $\lambda$ represents the degree of voluntary restraint under an outbreak. We set $\epsilon_k$ and $\sigma_k$ to be large and small, respectively, when job type $k$ is related to commodities, and vice versa when job type $k$ is related to non-essentials. Namely, demand for commodities increases constantly regardless of the economic state and the state of outbreak, while those for non-essentials are affected by the amount of money agents own and the degree of outbreak.

Agent $i$'s perception of the spread of infection $U_i$ is

$$U_i(t+1) = (1 - \kappa)U_i(t) + \kappa n_i(t), \tag{2.3}$$

where $n_i$ is the number of agents with state $Q$ within radius $r$ from agent $i$, which represents the number of patients monitored in the residential area. Here, we count only the number of quarantined agents based on the assumption that people cannot recognize that people with states $E$ and $I$ are actually infected. Equation (2.3) means that $U_i(t)$ increases or decreases, followed by an increase or decrease in $n_i(t)$. $U_i(t)$ is updated rapidly when $\kappa$ is large. Thus, $\kappa$ characterizes the delay between the actual and perceived disease spread (increase in hospitalized patients) around agent $i$; $\kappa$ increases as the delay decreases.

Each agent has a target list that represents the priority of goods to purchase (figure 1). When the demand $\rho_{i,k}$ exceeds the integral multiple of $\rho_{th}$, job type $k$ is added to the bottom of agent

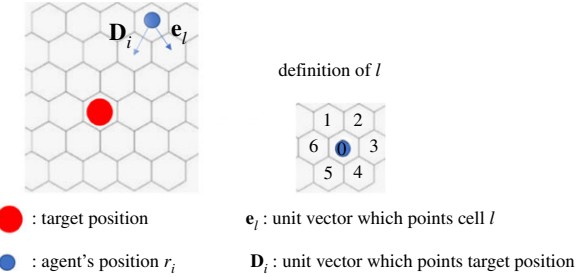

**Figure 3.** Definition of $D_i$ and $e_l$. (Online version in colour.)

$i$'s target list. Agent $i$ visits the home of the nearest agent, whose job type corresponds to the top of the target list. When the agent reaches its home, it pays a unit amount of money to buy the goods. Here, we simply assume that this deal holds even when the selling agent is not at home. Then because the demand of agent $i$ is satisfied, $\rho_{i,k}$ decreases by $\rho_{th}$. The top item in the target list is removed, and the other items in the target list move up. Then, agent $i$ visits the next target. When no items remain in the target list, agent $i$ returns to its home cell and remains there. Here, we assume that the order in the target list is never reversed for simplicity, although people often prioritize buying commodities when they are urgently needed.

We describe the rule for the movement of each agent as follows (figure 3). First, we define the target position of agent $i$ as the home of the nearest agent whose job type corresponds to the top of the target list. When no items remain in the target list, the target position is agent $i$'s home cell. We denote a unit vector that points the target position from the current position of agent $i$ by $D_i$. When the target position is identical to the current position, $D_i = 0$. Agent $i$ moves to its adjacent cells or stays in the current position with the probability $p_i(l)$:

$$p_i(l) = \frac{1}{6}(1 + e_l \cdot D_i), \ (l = 1, 2, 3, 4, 5, 6),$$
$$p_i(0) = 0,$$

(2.4)

in the case of $D_i \neq 0$, and

$$p_i(l) = 0, \quad (l = 1, 2, 3, 4, 5, 6),$$
$$p_i(0) = 1,$$

(2.5)

in the case of $D_i = 0$, where figure 3 provides the definitions of the unit vector $e_l$ and the subscript $l$. The term $e_l \cdot D_i$ is positive when cell $l$ is located near the direction of the target position and negative vice versa. Thus, owing to this term, agent $i$ tends to approach the target position.

## 3. Simulation results

We performed simulations of the proposed model. The number of cells along the horizontal and vertical directions are 43 and 50, respectively, and we adopt the periodic boundary condition. Except for the experiment illustrated in figure 4, we set the total number of agents $N$ to 1000. We set the initial amount of money to $\tilde{M}$ for all agents. We placed the home cells of the agents randomly without any overlap. Each agent is initially located at its home. We performed each trial for 30 000 time steps.

The simulation contains four job types, with an identical number of people for each job type. We determine the parameters $\epsilon_k$ and $\sigma_k$ ($k = 1, 2, 3, 4$) as

$$\left. \begin{array}{l} \epsilon_k = 0.1(k - 1) \\ \sigma_k = \dfrac{0.5 - \epsilon_k}{\tilde{M}} \end{array} \right\}.$$

(3.1)

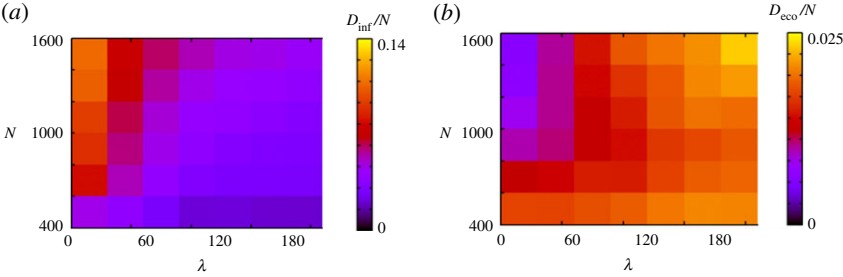

**Figure 4.** Simulation results when $N$ and $\lambda$ change. (a) and (b) show $D_{inf}/N$ and $D_{eco}/N$, respectively. (Online version in colour.)

**Table 1.** Simulation parameter values.

| parameter | value | parameter | value |
|---|---|---|---|
| $N$ | 1000 | $R$ | 10 |
| $p_E$ | 0.012 | $p_D$ | 0.0001 |
| $p_Q$ | 0.005 | $\kappa$ | 0.0004 |
| $\tau_E$ | 600 | $\tau_R$ | 1200 |
| $\rho_{th}$ | 100.0 | $K$ | 0.0007 |
| $\tilde{M}$ | 60 | $\hat{M}$ | 60 |

Thus, people with a high job type number sell commodities while those with a small job type number sell non-essentials. From equations (2.2) and (3.1), the rate of increase in demand $\rho_{i,k}$ does not depend on the job type when $M_i = \tilde{M}$.

We used non-dimensional parameters in the simulation. Because the proposed model is highly abstract, it is difficult to determine the parameters based on realistic data. However, we chose the duration of the latent period $\tau_E$ and that of infection $\tau_R$ to roughly mimic the properties of COVID-19. Specifically, we set $\tau_E$ and $\tau_R$ to 600 and 1200, respectively. These non-dimensional values correspond to 7 and 14 days, respectively, if each time step is rescaled to 16.8 min, and these values roughly agree with the property of COVID-19. We set the death probability $p_D$ such that the death rate becomes approximately 10% without voluntary restraint; that is, $\lambda = 0$. We determined the economic parameters such as $\tilde{M}, \hat{M}, \rho_{th}$ and $K$ such that death caused by the economic loss becomes almost zero in the absence of an outbreak, but some people die due to economic loss under voluntary restraint during an outbreak. Table 1 summarizes the parameter values, which we use unless otherwise specified hereafter.

## (a) Results when no patient exists

To capture the basic property of the proposed model, we performed the simulations in the no patient case; that is, $State_i = S$ holds for all $i$. By removing the factor of infection from the model, we can easily understand the basic property of economic activity. In this experiment, we changed the parameter $K$ to investigate the effect of income redistribution by taxes and public assistance. We evaluate the results in terms of the number of deaths caused by economic loss $D_{eco}$ and the variance of the amount of money $V[M] \equiv N^{-1} \sum_{i=1}^{N}(M_i - \bar{M})^2$, where $\bar{M}$ is the average $M_i$.

Supplementary videos 1–3 show the results when $K = 0, 0.00007$ and 0.00014, respectively. Figure 5 shows the time evolution of $V[M]$ and $D_{eco}$ for several values of $K$. We find that $V[M]$ increases over time. The rate of increase decreases as the value of $K$ increases (figure 5a). The number of deaths $D_{eco}$ increases over time for a small $K$, while it remains zero for a large $K$ (figure 5b). This result is reasonable because frequent buying and selling events cause random walk-like behaviour in $M_i$ and make the distribution of $M_i$ spread in a diffusive manner. The

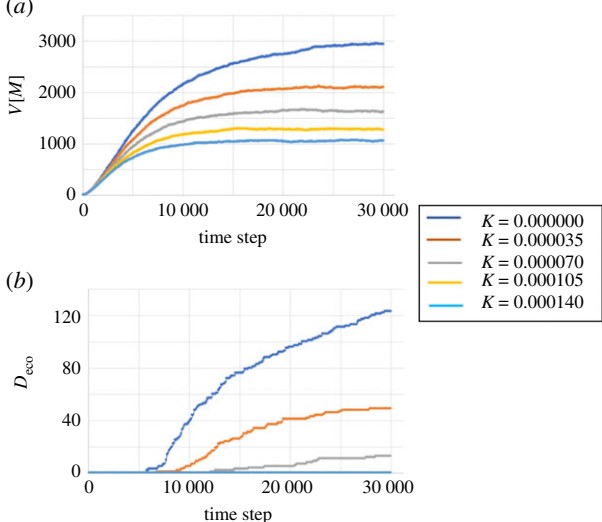

**Figure 5.** Results when no patient exists. The figure shows the time evolutions of (a) $V[M]$ and (b) $D_{eco}$ for several values of $K$. In (b), the lines for $K = 0.000105$ and $0.000140$ overlap. (Online version in colour.)

fourth term on the right-hand side of equation (2.1) suppresses the spread of the distribution of $M_i$. Thus, as $K$ increases, the difference between the rich and the poor decreases, and the number of deaths caused by economic loss decreases.

## (b) Effects of voluntary restraint

We performed simulations for the case in which the health state changes from $S$ to $E$ with the probability of 0.005 at time step 5000. We set parameter $K$ to 0.00007. We also simulated cases of $\lambda = 0$, 120 and 210 to investigate the effect of voluntary restraint.

Supplementary videos 4–6 show the results when $\lambda = 0$, 120 and 210, respectively. Figures 6a–c show the time evolution of the number of agents with each health state. When $\lambda = 0$, the infection spreads rapidly and more than 80% of agents are infected. The maximum number of quarantined patients (state $Q$) is 278, and the number of deaths caused by infection $D_{inf}(t)$ finally reached 101 (figure 6a). As $\lambda$ increases, the number of patients and the deaths caused by infection decrease considerably (figures 6b,c). When $\lambda = 210$, less than 20% of the agents were infected (figure 6c).

Next, we observe the economic tendencies in these cases. We evaluate the results by the total number of deaths caused by economic loss and the total amount of money for each job type, denoted by $D_{eco,k}(t)$ and $T_k(t)$, respectively ($k = 1, 2, 3, 4$). Namely, $T_k(t)$ is given by

$$T_k(t) = \sum_{i \in \text{job type } k} M_i(t). \tag{3.2}$$

Figures 6d–f and g–h, respectively, illustrate the time evolution of $D_{eco,k}(t)$ and $T_k(t)$ for the cases of $\lambda = 0$, 120 and 210. For $\lambda = 0$, only a few agents died because of economic loss (figure 6d). The total amount of money $T_k(t)$ remains almost constant (figure 6g). Voluntary restraint, that is, $\lambda > 0$, generates an economic gap between job types. Specifically, $T_k(t)$ becomes small for small $k$ and vice versa for large $k$ followed by the outbreak. The economic gap is the highest around 15 000 time steps, and it is mitigated gradually until around 30 000 time steps (figures 6h,i). The number of deaths caused by economic loss is larger for small $k$ (figure 6e), and it increases as $\lambda$ increases (figure 6f). This result suggests that voluntary restraint causes agents who sell non-essentials to suffer an economic loss, while those who sell commodities do not. However, the agents suffering from an economic loss can return to their normal lives after the outbreak is over.

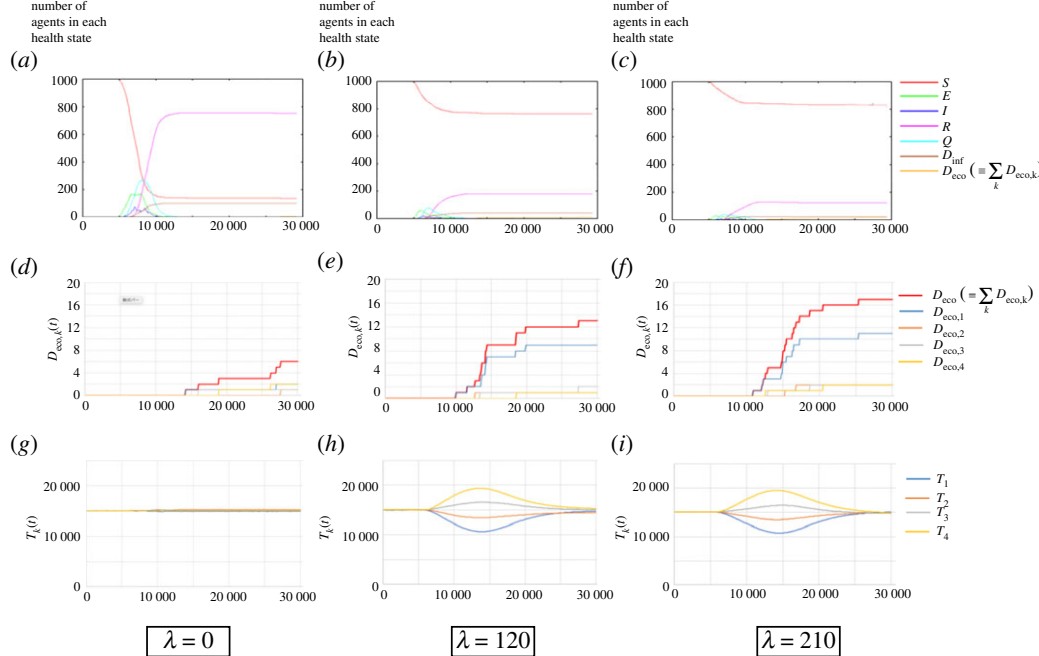

**Figure 6.** Simulation results when patients appear at time step 5000: ($a$)–($c$) Time evolution of the number of agents in each health state ($S, E, I, R, Q, D_\text{inf}$ and $D_\text{eco}$); ($d$)–($g$) time evolution of $D_{eco,k}(t)$ and $D_\text{eco}(\equiv \sum_k D_{eco,k}(t))$; and ($h$)–($i$) time evolution of $T_k(t)$. $\lambda = 0$ for ($a$), ($d$) and ($g$); 120 for ($b$), ($e$) and ($h$); and 210 for ($c$), ($f$) and ($i$). (Online version in colour.)

In summary, voluntary restraint is effective for mitigating an outbreak, although it induces an economic gap and increases the number of deaths caused by economic loss.

## (c) Effect of other parameters

To further clarify the properties of the proposed model, we performed simulations by changing several parameters expected to affect the resultant behaviours. Specifically, we changed the infectious rate $p_E$, the duration of infection $\tau_E$ and $\tau_R$, the amount of money $\tilde{M}$ and $\hat{M}$, the total number of agents $N$, and the speed of the response to the outbreak $\kappa$, since we expect that these parameters would affect the infection and economic dynamics. For each parameter, we also changed $\lambda$ and made colour maps for ease of analysis. The other simulation conditions were the same as those described in the previous subsection. We evaluate the results by the ratio of the number of deaths caused by infection $D_\text{inf}$ to the number of agents $N$ and that of the number of deaths caused by economic loss $D_\text{eco}$ to the number of agents $N$ at the 30 000th time step. We performed 10 trials for each parameter and report the average values of $D_\text{inf}/N$ and $D_\text{eco}/N$ hereafter. In the following subsections, we report the results when the parameters vary.

### (i) $p_E$ dependence

Figure 7 shows the result when we change the infectious rate $p_E$. For small $p_E$, $D_\text{inf}/N$ decreases as $\lambda$ increases, while $D_\text{eco}/N$ is almost unchanged depending on $\lambda$. Meanwhile, when $p_E$ is large, $D_\text{inf}/N$ is large for small $\lambda$, while $D_\text{eco}/N$ is large for large $\lambda$. This trade-off increases in severity as the infectious rate increases. This result suggests that when the infectious rate is low, voluntary restraint can stop the outbreak before economic loss becomes severe. When the infectious rate is high, infection spreads in the absence of voluntary restraint, and even strong voluntary restraint cannot completely stop the outbreak; thus, economic loss becomes severe under strong voluntary restraint.

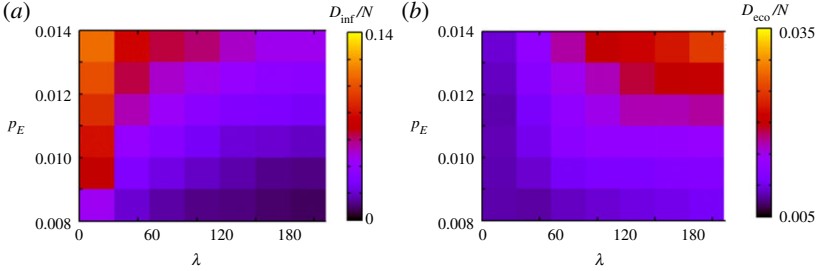

**Figure 7.** Simulation results when $p_E$ and $\lambda$ change. ($a$) and ($b$) show $D_{\text{inf}}/N$ and $D_{\text{eco}}/N$, respectively. (Online version in colour.)

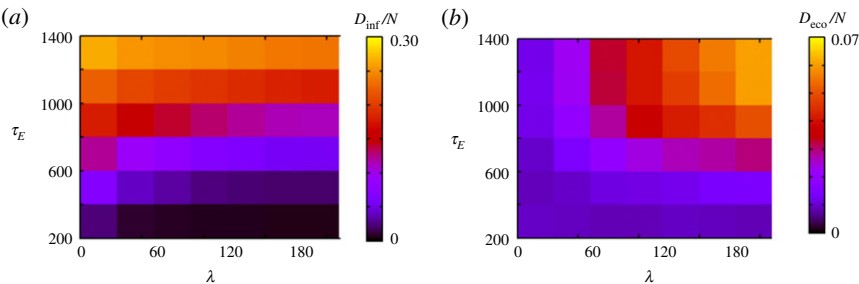

**Figure 8.** Simulation results when changing $\tau_E$, $\tau_R$ and $\lambda$ while satisfying $\tau_R = 2\tau_E$. ($a$) and ($b$) show $D_{\text{inf}}/N$ and $D_{\text{eco}}/N$, respectively. (Online version in colour.)

### (ii) $\tau_E$ and $\tau_R$ dependence

Figure 8 shows the result when we change the duration of infection $\tau_E$ and $\tau_R$ while maintaining the relation $\tau_R = 2\tau_E$. When $\tau_E$ and $\tau_R$ are small, $D_{\text{inf}}/N$ decreases as $\lambda$ increases, while $D_{\text{eco}}/N$ does not depend much on $\lambda$. When $\tau_E$ and $\tau_R$ are large, $D_{\text{inf}}/N$ is large for small $\lambda$, while $D_{\text{eco}}/N$ is large for large $\lambda$. Thus, the above-mentioned trade-off increases in severity as the duration of infection increases. This result suggests that the increase in $\tau_E$ and $\tau_R$ has an effect similar to the increase in $p_E$. When the duration of infection is short, voluntary restraint enables the cessation of the outbreak before economic loss becomes severe. However, it is not possible as the duration increases.

### (iii) $\tilde{M}$ and $\hat{M}$ dependence

Figure 9 shows the result when we change the initial amount of money $\tilde{M}$. We also change $\hat{M}$ in equation (2.1) such that $\hat{M} = \tilde{M}$ holds. While $D_{\text{inf}}/N$ decreases as $\lambda$ increases, it does not depend much on $\tilde{M}$. In contrast, $D_{\text{eco}}/N$ increases as $\tilde{M}$ decreases. Thus, when people have enough money, they can continue voluntary restraint until the outbreak ends. However, when people do not have enough money, voluntary restraint leads to an increase in death caused by economic loss. Consequently, people cannot undergo severe voluntary restraint, which leads to an increase in the death caused by infection.

### (iv) $N$ dependence

Figure 4 shows the result when we change the total number of agents $N$. When $N$ is large, $D_{\text{inf}}/N$ is large for a small $\lambda$ and $D_{\text{eco}}/N$ is large for a large $\lambda$. This is reasonable because when $N$ is large, people have many opportunities to meet others, and thus the probability of infection is high. When people adopt voluntary restraint, the death caused by economic loss increases. When $N$ is small, $D_{\text{inf}}/N$ is generally small, and $D_{\text{eco}}/N$ is not significantly affected by $\lambda$. In this case, few

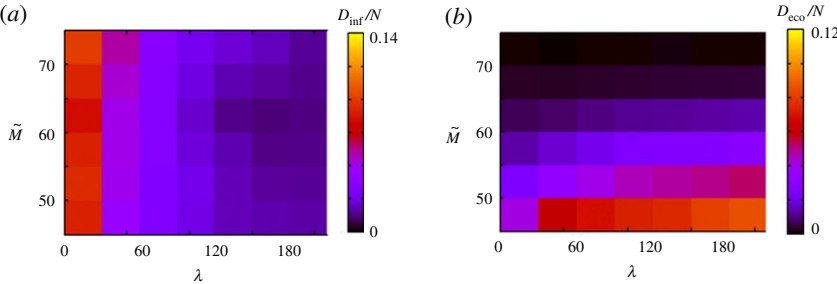

**Figure 9.** Simulation results when $\tilde{M}$ and $\lambda$ change. ($a$) and ($b$) show $D_{\text{inf}}/N$ and $D_{\text{eco}}/N$, respectively. (Online version in colour.)

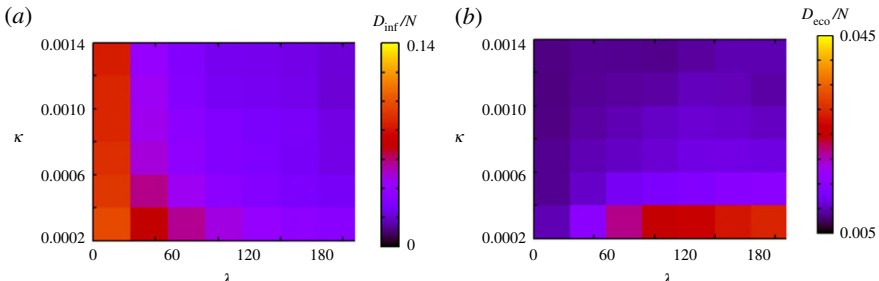

**Figure 10.** Simulation results when $\kappa$ and $\lambda$ change. ($a$) and ($b$) show $D_{\text{inf}}/N$ and $D_{\text{eco}}/N$, respectively. (Online version in colour.)

people are infected because they have few opportunities to meet others. The reason some people die due to economic loss is not related to infection. As figure 5$b$ ($K = 0.00007$) shows, a few people die due to economic loss even in the absence of infection.

### (v) $\kappa$ dependence

Figure 10 shows the result when we vary the speed of the response to the outbreak $\kappa$. When $\kappa$ is small, $D_{\text{inf}}/N$ tends to be larger than in the case of large $\kappa$, and $D_{\text{eco}}/N$ is large for a large $\lambda$. Meanwhile, when $\kappa$ is large, $D_{\text{eco}}/N$ decreases as $\lambda$ increases. This result suggests that the outbreak cannot be mitigated under a slow response against the spread of the virus. Under a fast response, voluntary restraint is effective because it can stop the outbreak before economic loss becomes severe. Thus, responding quickly is important for both mitigating outbreaks and maintaining economic activity.

## 4. Discussion and conclusion

In this paper, we proposed a simple and abstract mathematical model of a COVID-19 outbreak with economic activities. The simulation results show that voluntary restraint measures can help mitigate an outbreak, although it generates an economic gap between job types. We can explain the resulting economic gap as follows: the income of agents who sell non-essentials decreases during an outbreak because demand for non-essentials decreases owing to voluntary restraint. Since they must buy commodities as usual, they become poorer. By contrast, agents who sell commodities can get income as usual, yet they do not buy non-essentials due to voluntary restraint. Consequently, they become richer. This result suggests that in real societies, governments should make efforts to reduce the economic gap; otherwise, voluntary restraint measures cannot continue, which makes it difficult to mitigate an outbreak.

From figures 4 and 7–10, whether it is possible to mitigate outbreaks while minimizing economic loss depends on various factors such as the properties of the virus, people's economic levels and the speed of people's responses. The result in figure 9 suggests that people cannot tolerate voluntary restraint if their overall economic level is low. Thus, it is particularly difficult for communities with low economic power to mitigate outbreaks while maintaining economic activities. A possible solution to this problem is to respond to the spread of the virus as early as possible. As figure 10 suggests, if the response occurs earlier (which corresponds to a large $\kappa$ in our model), it is possible to stop an outbreak without causing severe economic damage. If people fail to respond quickly and the virus spreads, then people should anticipate other possibilities, such as the development of antiviral drugs and a weakening of the virus, which corresponds to decreasing $p_E$ in our model.

Because we did not consider production activities and simply assumed that the prime costs of goods are zero, our model is not suitable for making precise and quantitative predictions, especially on the economic impact. However, our model captures the essence of the interrelation between the spread of SARS-COV-2 and economic activities. Thus, we believe that our model can become a platform for discussing strategies to mitigate the outbreak while maintaining economic activities. Indeed, our model has many potential extensions. For example, it may be possible to account for the effect of long-distance movement with transportation modes such as airplanes and trains. Defining the capacity of hospitals may help policymakers discuss how to avoid overwhelming hospitals. It may also be interesting to introduce the individual characteristics of each agent and discuss how selfish people affect the spread of the virus and economic activities.

Moreover, beyond the outbreak of COVID-19, our model may share a common mechanism with other systems that can survive under harsh circumstances with limited resources, such as bacterial biofilms [33] and communities of vampire bats [34]. Extracting the common mechanism may lead to the establishment of a general design principle for artificial systems with high survivability.

Data accessibility. The data including the source code and the supplementary videos are provided as electronic supplementary material.

Authors' contributions. T.K. and M.A. conceived of the work. T.K. and K.Y. proposed the mathematical model, and T.M., M.A. and A.I. refined it. T.K. performed the simulations. All authors analysed the results and drafted the manuscript. All authors gave final approval for publication and agree to be held accountable for the work performed therein.

Competing interests. We declare we have no competing interests.

Funding. This work is in part supported by the JSPS KAKENHI Fund for the Promotion of Joint International Research (Fostering Joint International Research (B))19KK0103.

Acknowledgements. The authors would like to thank Dr Akira Fukuhara and Shura Suzuki of the Research Institute of Electrical Communication, Tohoku University for their helpful suggestions. The authors would like to thank Dr Masahiro Shimizu of Osaka University for providing part of the source code.

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
