## [Reviewer comments · Proceedings. Mathematical, Physical, and Engineering Sciences]

Review History

RSPA-2020-0604.R0 (Original submission)

Review form: Referee 1

Is the manuscript an original and important contribution to its field?

Good

Is the paper of sufficient general interest?

Good

Is the overall quality of the paper suitable?

Good

Do you have any ethical concerns with this paper?

No

Recommendation?

Accept with minor revision (please list in comments)

Comments to the Author(s)

In this paper, the authors propose an abstract agent-based model of the COVID-19 outbreak by considering economic activities. They have reported that the lockdown measures can mitigate an outbreak and it can also generates an economic gat between job types. Based on the simulaton

results, they proposed that narrowing the economic gap is extremely important to the sustainability of the lockdown strategy. The paper seemed interesting to me; however, it must be carefully revised before being published and the following points should be considered:

1) I have some problems with the model definition. For instance, The authors assume that agents with states E or I have infect ability because asymptomatic COVID-19 patients have infectability (L27 P5) . But the asymptomatic infected agents and the exposed agents are two completely different concepts. And the asymptomatic patients among the exposed population should be in the minority according to the current detection data. In fact, the exposed state may not be an independent state, e.g., susceptible agents and infectious agents can be also exposed. Therefore, I think the author should clearly define the connotation of these six different states and the relationship between them (S, E, I, Q, R, D).

2) Regarding Equation 2.1, the authors mentioned that the fourth term on the righthand side represents the effect of the income redistribution. But they didn't explain the meaning of K and M^{\wedge} in details, and how they can affect the income redistribution or why choose this formation. I also have serious reservation with the rest of some other Equations and parameters. The authors should describe their connotations in more detail and clearly.

3) Regarding the intro, the authors should branch out and acknowledge some excellent work published recently. For example:

All together to fight COVID-19, *Am. J. Trop. Med. Hyg.* 102, 1181-1183 (2020);

Forecasting COVID-19, *Front. Phys.* 8, 127 (2020);

Early spread of COVID-19 in Romania: Imported cases from Italy and human-to-human transmission networks, *R. Soc. Open Sci.* 7, 200780 (2020).

4) In Figs.6-10, the authors just described the patterns briefly and simply. It is recommended to explain the reasons for the corresponding results in combination with the dynamic mechanism of the model proposed in this MS. They can also supply some discussions in more depth at the last Section (the Discussion and Conclusion).

Review form: Referee 2

Is the manuscript an original and important contribution to its field?

Good

Is the paper of sufficient general interest?

Good

Is the overall quality of the paper suitable?

Acceptable

Can the paper be shortened without overall detriment to the main message?

Yes

Do you think some of the material would be more appropriate as an electronic appendix?

No

Do you have any ethical concerns with this paper?

No

Recommendation?

Accept with minor revision (please list in comments)

Comments to the Author(s)

Comment to the authors is given in the attached file.

Decision letter (RSPA-2020-0604.R0)

15-Oct-2020

Dear Dr Kano

The Editor of Proceedings A has now received comments from referees on the above paper and would like you to revise it in accordance with their suggestions which can be found below (not including confidential reports to the Editor).

Please consider the reviewers' comments and improve the readability. Reviewer 1's questions are useful in that they show that more explanation is needed for the general scientific audience of the Proceedings. The simple model does seem to achieve your aims. However ultra-discrete models such as cellular automata have some limitations. It would pay to mention some of those in the conclusion.

Please submit a copy of your revised paper within four weeks - if we do not hear from you within this time then it will be assumed that the paper has been withdrawn. In exceptional circumstances, extensions may be possible if agreed with the Editorial Office in advance.

Please note that it is the editorial policy of Proceedings A to offer authors one round of revision in which to address changes requested by referees. If the revisions are not considered satisfactory by the Editor, then the paper will be rejected, and not considered further for publication by the journal. In the event that the author chooses not to address a referee's comments, and no scientific justification is included in their cover letter for this omission, it is at the discretion of the Editor whether to continue considering the manuscript.

- Acknowledgements
- Funding statement

To revise your manuscript, log into <http://mc.manuscriptcentral.com/prsa> and enter your Author Centre, where you will find your manuscript title listed under "Manuscripts with Decisions." Under "Actions," click on "Create a Revision." Your manuscript number has been appended to denote a revision.

You will be unable to make your revisions on the originally submitted version of the manuscript. Instead, revise your manuscript and upload a new version through your Author Centre.

When submitting your revised manuscript, you will be able to respond to the comments made by the referee(s) and upload a file "Response to Referees" in "Section 6 - File Upload". Please use this to document how you have responded to the comments, and the adjustments you have made. In order to expedite the processing of the revised manuscript, please be as specific as possible in your response to the referee(s).

IMPORTANT: Your original files are available to you when you upload your revised manuscript. Please delete any unnecessary previous files before uploading your revised version.

When revising your paper please ensure that it remains under 28 pages long. In addition, any pages over 20 will be subject to a charge (£150 + VAT (where applicable) per page). Your paper has been ESTIMATED to be 12 pages.

Once again, thank you for submitting your manuscript to Proc. R. Soc. A and I look forward to receiving your revision. If you have any questions at all, please do not hesitate to get in touch.

Yours sincerely
Raminder Shergill
proceedingsa@royalsociety.org

on behalf of
Professor Matjaz Perc
Board Member
Proceedings A

Reviewer(s)' Comments to Author:
Referee: 1
Comments to the Author(s)

In this paper, the authors propose an abstract agent-based model of the COVID-19 outbreak by considering economic activities. They have reported that the lockdown measures can mitigate an outbreak and it can also generate an economic gap between job types. Based on the simulation results, they proposed that narrowing the economic gap is extremely important to the sustainability of the lockdown strategy. The paper seemed interesting to me; however, it must be carefully revised before being published and the following points should be considered:

1) I have some problems with the model definition. For instance, The authors assume that agents with states E or I have infect ability because asymptomatic COVID-19 patients have infectability (L27 P5) . But the asymptomatic infected agents and the exposed agents are two completely different concepts. And the asymptomatic patients among the exposed population should be in the minority according to the current detection data. In fact, the exposed state may not be an independent state, e.g., susceptible agents and infectious agents can be also exposed. Therefore, I think the author should clearly define the connotation of these six different states and the relationship between them (S, E, I, Q, R, D).

2) Regarding Equation 2.1, the authors mentioned that the fourth term on the righthand side represents the effect of the income redistribution. But they didn't explain the meaning of K and M^{\wedge} in details, and how they can affect the income redistribution or why choose this formation. I also have serious reservation with the rest of some other Equations and parameters. The authors should describe their connotations in more detail and clearly.

3) Regarding the intro, the authors should branch out and acknowledge some excellent work published recently. For example:
All together to fight COVID-19, *Am. J. Trop. Med. Hyg.* 102, 1181-1183 (2020);
Forecasting COVID-19, *Front. Phys.* 8, 127 (2020);
Early spread of COVID-19 in Romania: Imported cases from Italy and human-to-human transmission networks, *R. Soc. Open Sci.* 7, 200780 (2020).

4) In Figs.6-10, the authors just described the patterns briefly and simply. It is recommended to explain the reasons for the corresponding results in combination with the dynamic mechanism of the model proposed in this MS. They can also supply some discussions in more depth at the last Section (the Discussion and Conclusion).

Referee: 2
Comments to the Author(s)
Comment to the authors is given in the attached file.

Board Member:
Comments to Author(s):
Based on the advice received by our referees, we will be happy to consider a revised manuscript that takes the comments into account.

RSPA-2020-0604.R1 (Revision)

Review form: Referee 1

Is the manuscript an original and important contribution to its field?

Good

Is the paper of sufficient general interest?

Good

Is the overall quality of the paper suitable?

Good

Can the paper be shortened without overall detriment to the main message?

Yes

Do you think some of the material would be more appropriate as an electronic appendix?

No

Do you have any ethical concerns with this paper?

No

Recommendation?

Accept as is

Comments to the Author(s)

The authors have examined all the comments thoroughly and have made corrections so that they meet with the approval. In general, the revised version meets the requirements of RSPA and I recommended for publication.

Review form: Referee 2

Is the manuscript an original and important contribution to its field?

Good

Is the paper of sufficient general interest?

Good

Is the overall quality of the paper suitable?

Good

Can the paper be shortened without overall detriment to the main message?

Yes

Do you think some of the material would be more appropriate as an electronic appendix?

Yes

Do you have any ethical concerns with this paper?

No

Recommendation?

Accept as is

Comments to the Author(s)

Your revision is good enough.

Decision letter (RSPA-2020-0604.R1)

03-Dec-2020

Dear Dr Kano

I am pleased to inform you that your manuscript entitled "An agent-based model of the interrelation between the COVID-19 outbreak and economic activities" has been accepted in its final form for publication in Proceedings A.

Our Production Office will be in contact with you in due course. You can expect to receive a proof of your article soon. Please contact the office to let us know if you are likely to be away from e-mail in the near future. If you do not notify us and comments are not received within 5 days of sending the proof, we may publish the paper as it stands.

COVID-19 rapid publication process: We are taking steps to expedite the publication of research relevant to the pandemic. If you wish, you can opt to have your paper published as soon as it is ready, rather than waiting for it to be published on the scheduled Wednesday. This means your paper will not be included in the weekly media round-up which the Society sends to journalists ahead of publication. However, it will appear in the COVID-19 Publishing Collection which journalists will be directed to each week

(<https://royalsocietypublishing.org/topic/special-collections/novel-coronavirus-outbreak>)

If you wish to have your paper published immediately please notify proca_proofs@royalsociety.org and press@royalsociety.org

Open access

Thank you for opting for open access for your paper. The Royal Society has signed a Wellcome statement on the subject of research findings and data relevant to the coronavirus (covid-19) outbreak. We are one of several signatories to this statement and our collective aim is to ensure that the relevant research and data are shared rapidly and openly in order to inform the worldwide public health response and to help save lives. We are therefore making papers related to covid-19 open access free of charge so you will not be invoiced for the open access fee.

Under the terms of our licence to publish you may post the author generated postprint (ie. your accepted version not the final typeset version) of your manuscript at any time and this can be made freely available. Postprints can be deposited on a personal or institutional website, or a recognised server/repository. Please note however, that the reporting of postprints is subject to a

media embargo, and that the status the manuscript should be made clear. Upon publication of the definitive version on the publisher's site, full details and a link should be added.

You can cite the article in advance of publication using its DOI. The DOI will take the form: 10.1098/rspa.XXXX.YYYY, where XXXX and YYYY are the last 8 digits of your manuscript number (eg. if your manuscript number is RSPA-2017-1234 the DOI would be 10.1098/rspa.2017.1234).

For tips on promoting your accepted paper see our blog post:
<https://royalsociety.org/blog/2020/07/promoting-your-latest-paper-and-tracking-your-results/>

On behalf of the Editor of Proceedings A, we look forward to your continued contributions to the Journal.

Sincerely,
Raminder Shergill
proceedingsa@royalsociety.org

Reviewer(s)' Comments to Author:
Referee: 1

Comments to the Author(s)
The authors have examined all the comments thoroughly and have made corrections so that they meet with the approval. In general, the revised version meets the requirements of RSPA and I recommended for publication.

Referee: 2

Comments to the Author(s)
Your revision is good enough.